# Isolation and Characteristics of a Novel Aichivirus D from Yak

Nan Yan,[a,b] Hua Yue,[b] Quan Liu,[f] Gang Wang,[c,d,e] Cheng Tang,[b] Ming Liao[a,c,d,e]

aNational and Regional Joint Engineering Laboratory for Medicament of Zoonosis Prevention and Control, Guangdong Provincial Key Laboratory of Zoonosis Prevention and Control, College of Veterinary Medicine, South China Agricultural University, Guangzhou, China
bCollege of Animal & Veterinary Sciences, Southwest Minzu University, Chengdu, China
cKey Laboratory of Livestock Disease Prevention of Guangdong Province, Guangzhou, China
dField Observation and Experiment Station on Animal Blight of Guangdong Province, Guangzhou, China
eInstitute of Animal Health, Guangdong Academy of Agricultural Sciences, Guangzhou, China
fSchool of Life Sciences and Engineering, Foshan University, Foshan, China

Nan Yan and Hua Yue contributed equally to this article. Author order was determined on the basis of their contributions.

**ABSTRACT** Aichivirus D (AiV-D) is a newly emerging *Kobuvirus* detected in bovine and sheep, and information is limited regarding its biological significance and prevalence. This study aimed to explore both the prevalence and characteristics of AiV-D in yaks. From May to August 2021, 117 fecal samples were collected from yaks with diarrhea in three provinces of China's Qinghai-Tibet Plateau, 15 of which were selected and pooled for metagenomic analysis. A high abundance of AiV-D sequences was obtained. Of the 117 diarrhea samples, 29 (24.8%) tested AiV-D–positive, including 33.3% (14/42) from Sichuan, 21.1% (8/38) from Qinghai, and 18.9% (7/37) from Tibet, respectively, suggesting a wide geographical distribution of the AiV-D in yaks in the Qinghai-Tibet Plateau. Furthermore, three AiV-D strains were successfully isolated using Vero cells. Significantly, the AiV-D strain could cause diarrhea, intestinal bleeding, and inflammation in yak calves via oral inoculation. The virus was distributed in the ileum, jejunum, duodenum, colon, cecum, and rectum. Based on phylogenetic analysis of the genome and capsid protein P1 (VP0, VP3, and VP1 genes), the yak AiV-D strains likely represent a novel genotype of AiV-D. On the whole, this study identified a novel genotype of AiV-D from yaks, which was successfully isolated, and confirmed that this virus is a diarrhea pathogen in yaks and has a wide geographical distribution in the Qinghai-Tibet Plateau. Our results expand the host range of AiV-D and the pathogen spectrum of yaks and have significant implications for diagnosing and controlling diarrhea in yaks.

**IMPORTANCE** In this study, we identified and successfully isolated a novel genotype of AiV-D from yaks. Animal infection confirmed that this virus can cause diarrhea, intestinal bleeding, and inflammation in yak calves via oral inoculation. The virus was distributed in the ileum, jejunum, cecum, duodenum, colon, and rectum. All of these results have significant implications for diagnosing and controlling diarrhea in yaks. These novel AiV-D strains have a wide geographical distribution in yaks from the Qinghai-Tibet Plateau in China. In addition to expanding the host range of AiV-D and the pathogen spectrum of yaks, these findings can increase knowledge of the prevalence and diversity of AiV-D.

**KEYWORDS** Aichivirus D, yak, diarrhea, isolation, pathogenicity

*K*obuvirus (KoV) is associated with diarrhea in humans and animals around the world (1). The *Kobuvirus* genome comprises a 5′ UTR (untranslated region), a large open reading frame (ORF), and a 3′ UTR. Moreover, the ORF encodes structural proteins, including P1 (VP0, VP3 and VP1), and nonstructural proteins, including P2 (2A-2C) and P3 (3A-3D) (2). Currently, there are six known *Kobuvirus* species (Aichivirus A to F): Aichivirus A can infect humans, dogs, cats, and birds (3–6); Aichivirus B has been found in cattle, sheep, and ferrets

Address correspondence to Cheng Tang, tangcheng101@163.com, or Ming Liao, mliao@scau.edu.cn.

The authors declare no conflict of interest.

**TABLE 1** Yak fecal sample collection and AiV-D detection[a]

| | Province and farm no. | | | | | | | |
|---|---|---|---|---|---|---|---|---|
| | Sichuan | | | Qinghai | | Tibet | | |
| Characteristic | 1 | 2 | 3 | 4 | 5 | 6 | 7 | Total |
| Samples, *n* | 16 | 12 | 14 | 17 | 21 | 17 | 20 | 117 |
| AiV-D positivity, % (*n*/total) | 31.3% (5/16) | 33.3% (4/12) | 28.6% (4/14) | 23.5% (4/17) | 19.0% (4/21) | 17.6% (3/17) | 20.0% (4/20) | 24.8% (29/117) |
| Farm positivity, % (*n*/total) | 100% (3/3) | | | 100% (2/2) | | 100% (2/2) | | 100% (7/7) |

[a]AiV-D, Aichivirus D.

(7–9); Aichivirus C consists of porcine and caprine kobuvirus (10, 11); and Aichivirus E and F contain rabbit and bat kobuvirus, respectively (12, 13).

In 2015, Aichivirus D (AiV-D) was first identified from diarrhea fecal samples of bovine in Japan (14). Recently, AiV-D was identified in sheep in China's Qinghai-Tibet Plateau (15). Currently, in cattle, AiV-D has only been detected in Japan and was divided into two genotypes (AiV-D1 and AiV-D2) by the *Kobuvirus* species classification criteria of ICTV (https://talk .ictvonline.org). In addition, the AiV-D from sheep likely represents a novel genotype of AiV-D (15). To date, three AiV-D genome sequences (two cattle and one sheep) are available in GenBank, and the nucleotide sequence identities between them range from 62.9% to 77.8%. Phylogenetic analysis has shown that these three AiV-D genomes may represent three distinct genotypes of AiV-D because they can be clustered into three independent branches in AiV-D (15). However, due to unsuccessful isolation of this virus and a lack of epidemiological data, the pathogenicity of AiV-D in cattle and sheep remains to be determined and its host range requires further investigation.

In recent years, advances in next-generation sequencing (NGS) technologies have resulted in the discovery of multiple divergent kobuviruses in various animals (16–18), and all of these sequences suggest that *Kobuvirus* have a high genetic diversity (16, 18). Yaks belong to the genus *Bos* in the family Bovidae and are a unique long-haired bovine species distributed in high-altitude regions (above 2,500 to 6,000 m) in China, Nepal, Kyrgyzstan, India, Pakistan, Mongolia, and Russia, but mainly in the Qinghai-Tibet Plateau in China (19). This research aims to explore the prevalence and characteristics of AiV-D in yaks.

## RESULTS

**High-throughput sequencing.** In our study, we performed deep sequencing and identified 4 distinct viruses in yak diarrhea samples (Fig. S1 in the supplemental material), listed in order of sequence read abundance: *Kobuvirus* (49.57% of all reads), *Astrovirus* (49.16%), *Enterovirus* (1.14%), and *Rotavirus* A (0.13%). A large sequence contig (7,686 bp) was assembled, which indicates the most significant similarity to the genome of AiV-D strain Kago-2-24 (GenBank no. LC055960), with 81.18% sequence identity.

**Detection of AiV-D in yaks.** Of the 117 diarrhea samples from yak, 29 (24.8%) were determined to be AiV-D–positive by quantitative real-time PCR (qRT-PCR), with positive rates of 33.3% (14/42), 21.1% (8/38), and 18.9% (7/37) in Sichuan, Qinghai, and Tibet, respectively. This result revealed a distribution of AiV-D in all 7 farms across 3 provinces (Table 1).

**Amplification of P1 region sequences.** To further investigate the molecular characteristics of the P1 region (VP0, VP3, and VP1 genes) of yak AiV-D strains, the complete P1 region sequence was obtained from the 14 AiV-D–positive samples (6 from Sichuan, 4 from Tibet, and 4 from Qinghai). All 29 AiV-D–positive samples were used for P1 region amplification; the P1 sequence was only successfully amplified from 14 AiV-D–positive samples, likely due to the low virus amounts in the clinical samples. All 14 P1 region sequences were 2,625 bp in length and encoded a total of 875 amino acid (aa) residues (GenBank no. OP776095 to OP776108). A phylogenetic tree was established following the 14 yak AiV-D complete P1 region sequences and the P1 region sequences of some other Aichivirus species in GenBank. These findings showed that the 14 P1 region sequences in yaks clustered into the AiV-D species but were located in an independent branch, showing no significant geographical differences (Fig. 1). In addition, in the phylogenetic trees based on the individual genes of the P1 region (VP0, VP3 and VP1 genes), it was also shown that

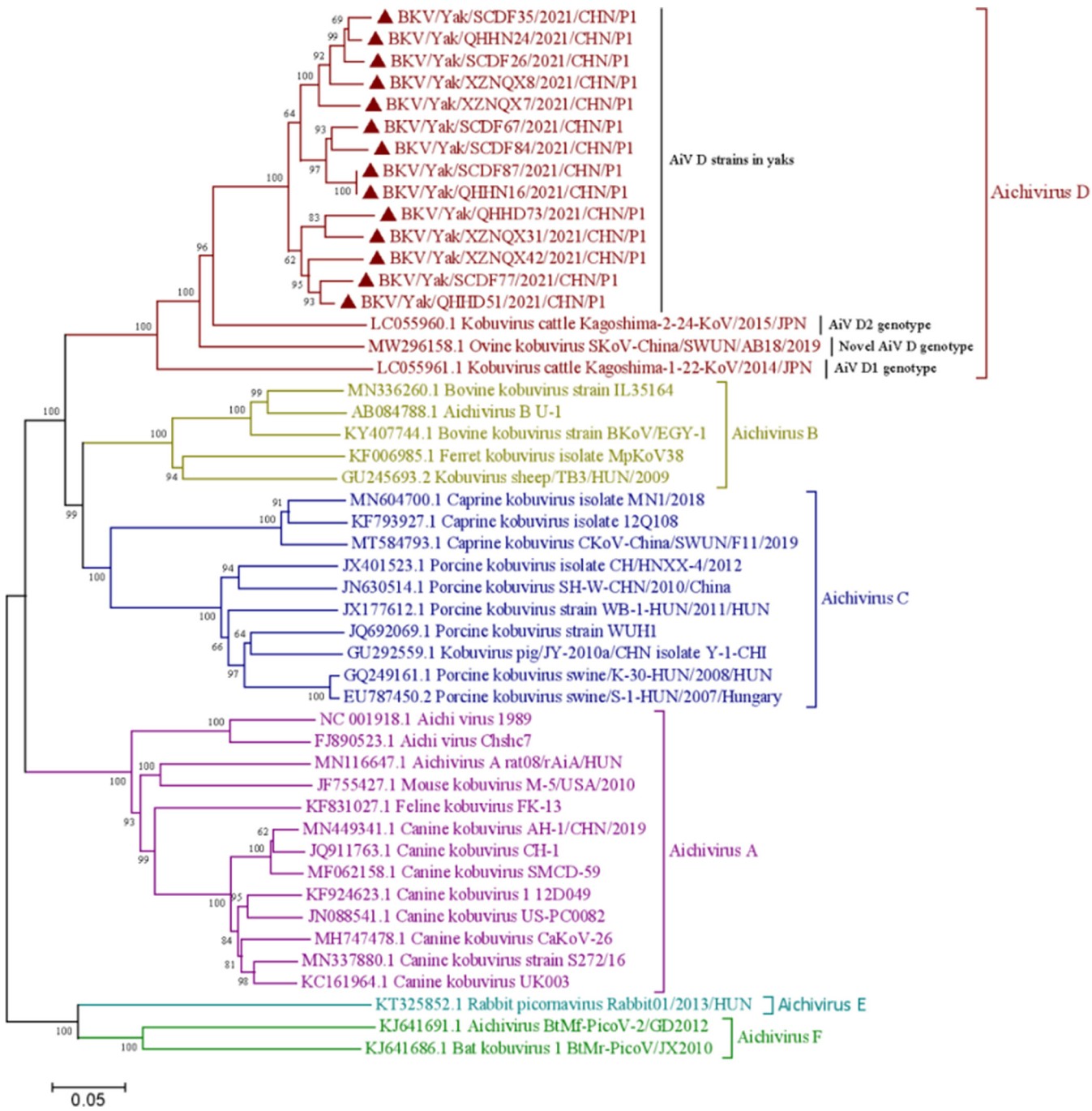

**FIG 1** Phylogenetic tree of Aichivirus (AiV) P1 protein nt sequences. Maximum-likelihood analysis in combination with 1,000 bootstrap replicates was used to derive a phylogenetic tree based on the complete nucleotide sequences of AiV P1 protein. Black triangles (▲) represent the AiV strains from this study.

these strains were clustered into AiV-D species and were located in an independent branch (Fig. S2 to S4). Additionally, compared with the known AiV-D strain sequences, there were 15 unique aa changes in the VP0 gene, 7 of which were located in the receptor-binding domain (RBD; 200 to 261 aa) of VP0. In the VP3 gene, 14 sequences in the research shared 13 aa changes; and in the VP1 gene, 31 unique aa changes were present, 3 of which (TV22I, A24E, and NMV43R) were located in the conserved antigenic peptide (22 to 44 aa) of VP1 (Fig. S5).

**Virus isolation and identification.** The AiV-D–positive samples were inoculated into the Vero cells. Of these 29 samples, only 3 exhibited obvious cytopathic effects (CPE) with the features of round shrinkage, shedding, and fusion from 48 to 72 h during the first

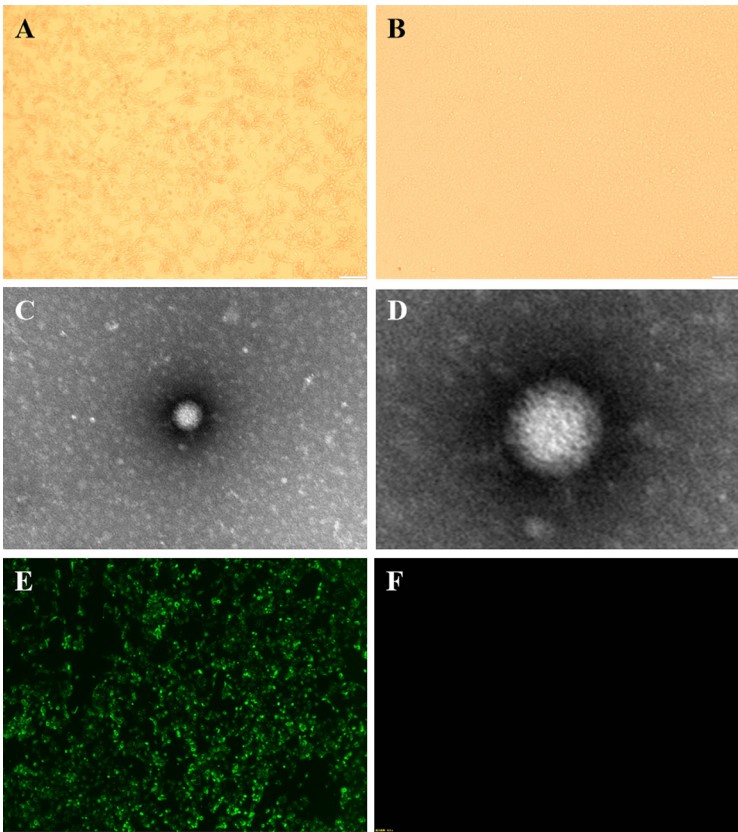

**FIG 2** Isolation and identification of AiV-D. (A) Vero cell infected with AiV-D strain showing cytopathic effects (CPE) at 72 h. (B) Mock-infection negative-control Vero cell. (C and D) TEM images of AiV-D strain. (E) Immunofluorescence (IF) of AiV-D strain. (F) IF negative control (mock-infected cells).

generation, and stable CPE were observed around 72 h after generation six (Fig. 2A). Three virus isolates were plaque-purified, and the virus titers were determined to be $10^{6.5}$, $10^{6.8}$, and $10^7$ median tissue culture infective dose ($TCID_{50}$)/0.1 mL for BKV/Yak/QHHN16/2021/CHN (Yak/HN16), BKV/Yak/XZNQX7/2021/CHN (Yak/NQX7), and BKV/Yak/SCDF77/2021/CHN (Yak/DF77), respectively. These isolates were verified to be KoV by immunofluorescence (IF) (Fig. 2E), and a spherical, nonenveloped virus particle (approximately 50 nm in diameter) was also visualized on a transmission electron microscope (TEM; Fig. 2C and D).

**Genomic characterization of AiV-D isolates in yaks.** Three complete genomes of the strains Yak/DF77 (GenBank no. OP776109), Yak/HN16 (OP776110), and Yak/NQX7 (OP776111) were obtained from the three isolates. These three genomes measure 8,484 (with 55.32% GC content), 8,484 (55.01% GC), and 8,477 nucleotides (nt) (55.14% GC) in length, respectively. Specifically, they possessed standard *Kobuvirus* genome organization: Yak/DF77 and Yak/HN16 contain a 7,512-bp complete ORF, and Yak/NQX7 contains a 7,506-bp complete ORF and encodes a polyprotein comprising structural proteins, including P1 (VP0, VP3, and VP1), and the nonstructural proteins P2 (2A to 2C) and P3 (3A to 3D). Structural maps of the three genomes are shown in Fig. 3. The three genome sequences shared 93.1% to 96.2% nt identity and shared the highest nt and aa identities with the AiV-D2 strain Kago-2-24. Table 2 shows the nt identity and aa identity of individual genes of Yak/DF77 compared with representative strains of six Aichivirus species, and Tables S2 to S3 show the other two genomes. In the phylogenetic trees based on the complete genomes, these strains were grouped into the AiV-D species, forming an independent branch (Fig. 4).

Moreover, the potential secondary structure of the 5′ UTR of the genomes included the three stem-loop domains SL-A, SL-B, and SL-C (Fig. S6). SL-A and SL-C were completely concordant among the three strains in this study; in SL-B, Yak/DF77 was identical to Yak/NQX7, but Yak/HN16 differed from the other two strains by 6 nt positions in SL-B. Further

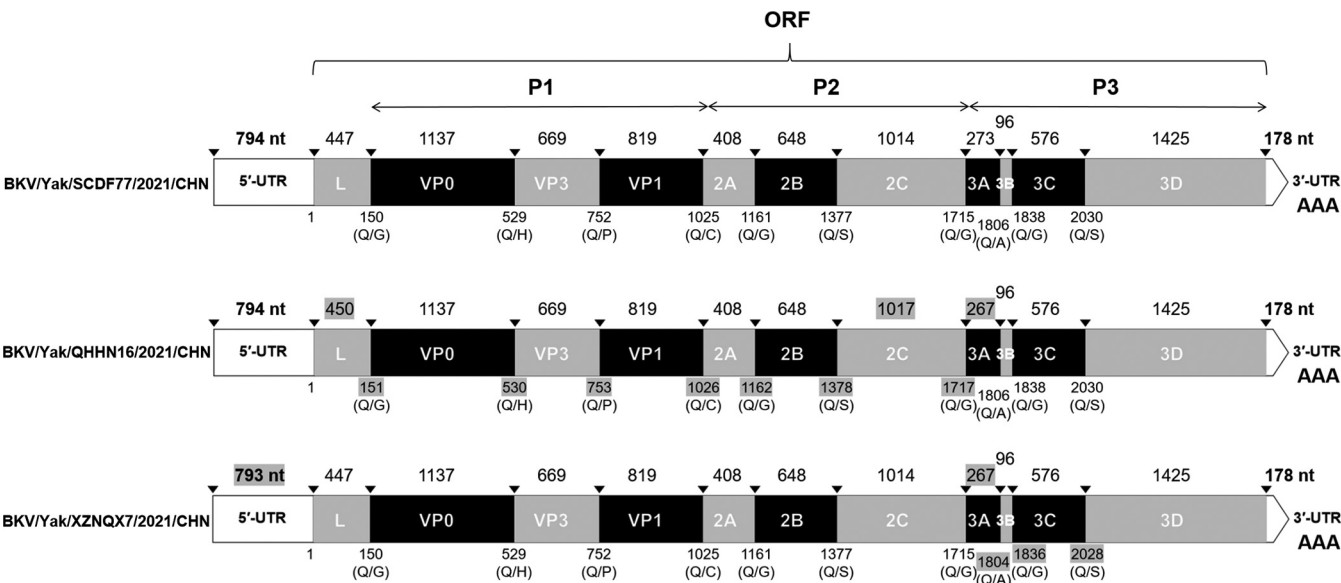

**FIG 3** Schematic representation of the yak AiV D genomes. Length of each gene is indicated by plain numbers. Differences in length between strains are indicated by gray background numbers. Black triangles (▼) show predicted cleavage sites.

analysis showed that the 5′ UTR conformations of these strains were different from those of known AiV-D strains. Specifically, in SL-A, three yak AiV-D strains differed from AiV-D2 geno-type strain Kago-2-24 at 5 nt positions and from novel genotype AiV-D strain AB18 at 21 nt positions; in SL-C, three yak AiV-D strains differed from AiV-D2 genotype strain Kago-2-24 at 7 nt positions and from novel genotype AiV-D strain AB18 at 5 nt positions; in SL-B, Yak/DF77 and Yak/NQX7 were identical to AiV-D2 genotype strain Kago-2-24 and differed from novel genotype AiV-D strain AB18 at 1 nt position; and finally, Yak/HN16 differed from AiV-D2 geno-type strain Kago-2-24 at 6 nt positions and from novel genotype AiV-D strain AB18 at 7 nt positions (Fig. S6). As a result, we found that the conformation of the 5′ UTR of AiV-D strains in yaks is different from those of the known AiV-D strains due to the existence of unique nt changes.

**Experimental infection of animals with AiV-D.** There was no clinical manifestation in infected BALB/c mice from days 1 to 7, and mice were euthanized 7 days postinfection (dpi). AiV-D was not detected by qRT-PCR in the infected mouse tissues (heart, cecum, colon,

**TABLE 2** BKV/Yak/SCDF77/2021/CHN nt and aa identity with other Aichivirus genome sequence[a]

| Gene region | Shared nt identity (%)/aa identity (%) | | | | | | | |
|---|---|---|---|---|---|---|---|---|
| | AiV-A Human/A846/88 | AiV-B Bovine/U-1 | AiV-C Caprine/F11 | AiV-D1 Kago-1-22 | AiV-D2 Kago-2-24 | AiV novel genotype D Ovine/AB18 | AiV-E Rabbit/01 | AiV-F Bat/BtMf-picov-2 |
| 5′ UTR | 32.9/- | 58.9/- | 45.8/- | - | 85.5/- | 85.9/- | 32.4/- | - |
| L | 42.5/**36.5** | 38.2/**34.7** | 33.8/**27.0** | 67.1/**61.3** | 79.1/**82.0** | 78.9/**79.3** | 28.9/**18.5** | 28.3/**17.6** |
| VP0 | 57.8/**50.4** | 59.0/**56.8** | 61.6/**60.4** | 72.1/**77.2** | 81.2/**89.5** | 78.5/**85.2** | 48.2/**37.1** | 52.9/**43.5** |
| VP3 | 61.8/**55.1** | 63.6/**64.5** | 62.9/**60.0** | 73.8/**80.4** | 79.7/**87.3** | 79.1/**85.7** | 58.5/**58.0** | 36.2/**21.0** |
| VP1 | 51.2/**39.4** | 57.6/**49.4** | 50.1/**44.6** | 64.1/**69.2** | 79.1/**86.9** | 78.5/**84.6** | 42.0/**29.8** | 29.7/**19.5** |
| 2A | 48.1/**46.6** | 58.4/**52.7** | 56.4/**51.4** | 69.0/**65.8** | 91.5/**93.8** | 80.0/**80.1** | 50.3/**41.1** | 34.8/**22.3** |
| 2B | 45.8/**38.9** | 41.9/**36.9** | 44.6/**39.3** | 46.9/**45.1** | 78.7/**83.8** | 74.7/**74.2** | 33.3/**26.4** | 37.6/**29.4** |
| 2C | 60.8/**54.2** | 59.8/**55.3** | 59.4/**55.8** | 65.8/**64.5** | 81.1/**85.6** | 79.2/**80.8** | 46.5/**20.2** | 36.9/**25.4** |
| 3A | 49.7/**39.0** | 45.0/**39.0** | 49.0/**43.8** | 59.3/**52.4** | 83.3/**88.6** | 72.0/**77.1** | 42.3/**30.5** | 29.0/**23.3** |
| 3B | 50.5/**34.3** | 54.4/**51.4** | 46.6/**40.0** | 53.4/**54.3** | 82.5/**88.6** | 66.0/**77.1** | 38.8/**28.6** | 39.8/**25.7** |
| 3C | 57.0/**50.7** | 54.5/**47.8** | 52.4/**45.3** | 64.1/**58.1** | 82.8/**87.2** | 79.0/**81.3** | 47.6/**32.0** | 34.3/**27.5** |
| 3D | 64.8/**65.5** | 64.7/**66.3** | 61.8/**63.9** | 73.2/**77.8** | 87.3/**92.2** | 79.8/**84.2** | 57.9/**55.3** | 57.8/**56.1** |
| 3′ UTR | 31.8/- | 57.4/- | 53.1/- | 77.3/- | 91.3/- | 93.1/- | 52.3/- | 25.6/- |
| Complete | 54.7/- | 55.6/- | 54.9/- | 64.9/- | 82.6/- | 79.2/- | 45.9/- | 43.7/- |
| ORF | 55.7/**49.7** | 56.0/**52.7** | 55.7/**52.3** | 66.9/**66.7** | 82.1/**86.8** | 78.1/**80.5** | 47.9/**41.2** | 49.1/**42.1** |

[a]nt, nucleotide; aa, amino acid; AiV, Aichivirus; UTR, untranslated region; ORF, open reading frame. aa identity shown in bold.

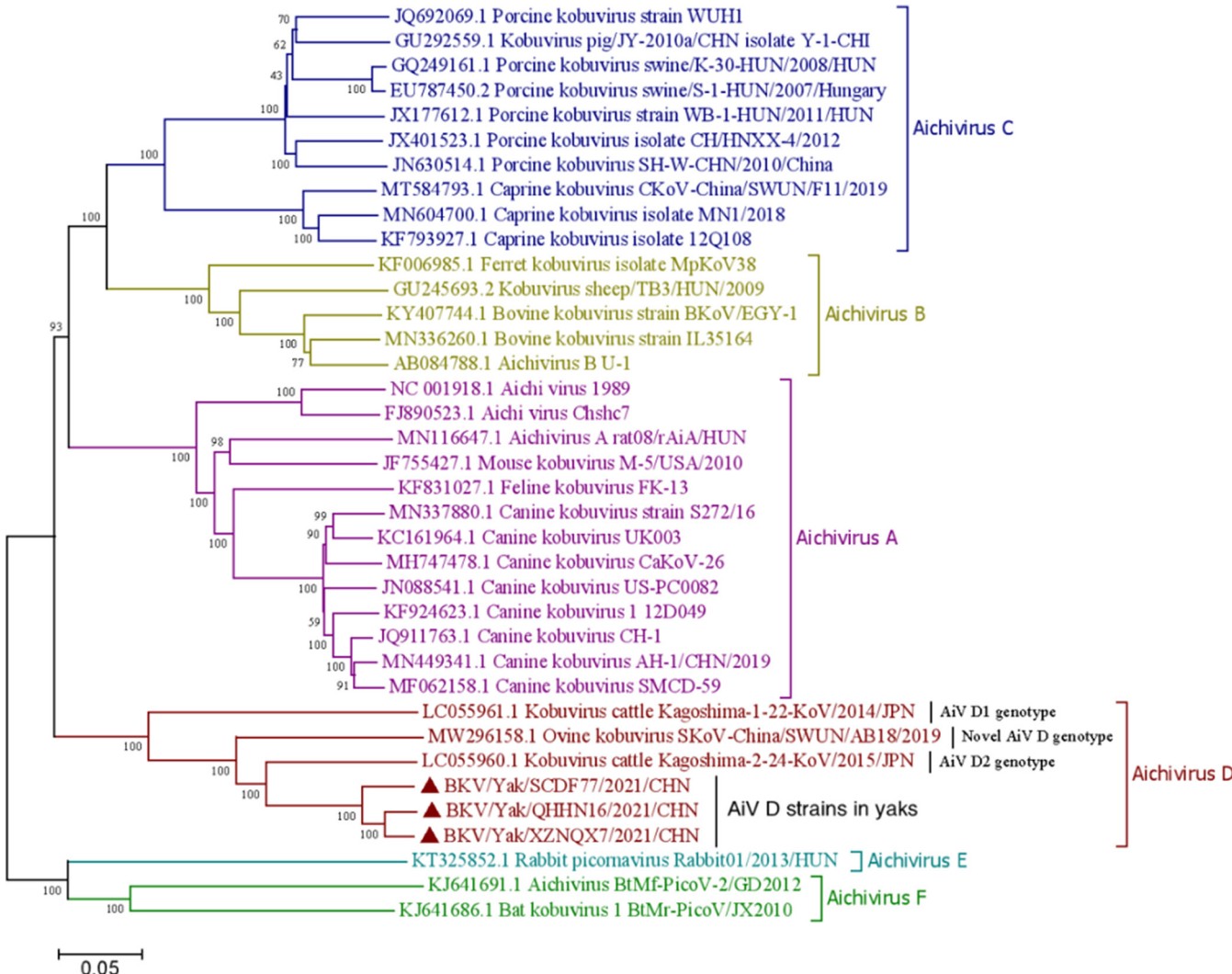

**FIG 4** Phylogenetic tree of the genomes of six species (A to F) of AiV. Maximum-likelihood analysis in combination with 1,000 bootstrap replicates was used to derive a phylogenetic tree based on the complete genome sequences of six AiV strains. Black triangles (▲) represent the AiV strains from this study.

liver, duodenum, lung, feces, kidney, jejunum, ileum, rectum, spleen, lymph nodes, and blood). These findings show that the AiV-D isolate did not successfully infect mice in this study.

The infected yak calves showed watery diarrhea and depression at 3 dpi, and diarrhea was the most severe at 6 dpi (Fig. S7). The two infected yak calves were euthanized at 6 and 8 dpi, and gross pathological changes were found within the duodenum and jejunum, featuring intestinal bleeding (Fig. S8). Histopathological changes were found in the intestine, including shedding of mucosal epithelial cells, necrosis of the intestinal glands, and hyperemia (Fig. 5A and B). KoV immunohistochemistry expression was also noted for intestinal tissues from infected yak calves, showing that the AiV-D virus was present (Fig. 5C). There were no clear pathological changes in the ileum, cecum, colon, and rectum, but hematoxylin and eosin (H&E) staining showed necrosis of the mucosal layer in these regions (Fig. S9). In the control yak calf, no clinical manifestations or macroscopic lesions were found. In this study, AiV-D was first determined to be pathogenic to yak calves, expanding the pathogen spectrum of yaks.

**Fecal shedding of AiV-D in yaks.** Anal swabs were collected from each yak calf on days 1 to 7, 9, 11, and 14 postinfection, and virus shedding in the calves was examined using qRT-PCR. AiV-D was detected in anal swabs from the infected calves from 1 to 11 dpi, and the virus was completely cleared by 14 dpi. The amount of virus shedding peaked from 5 to

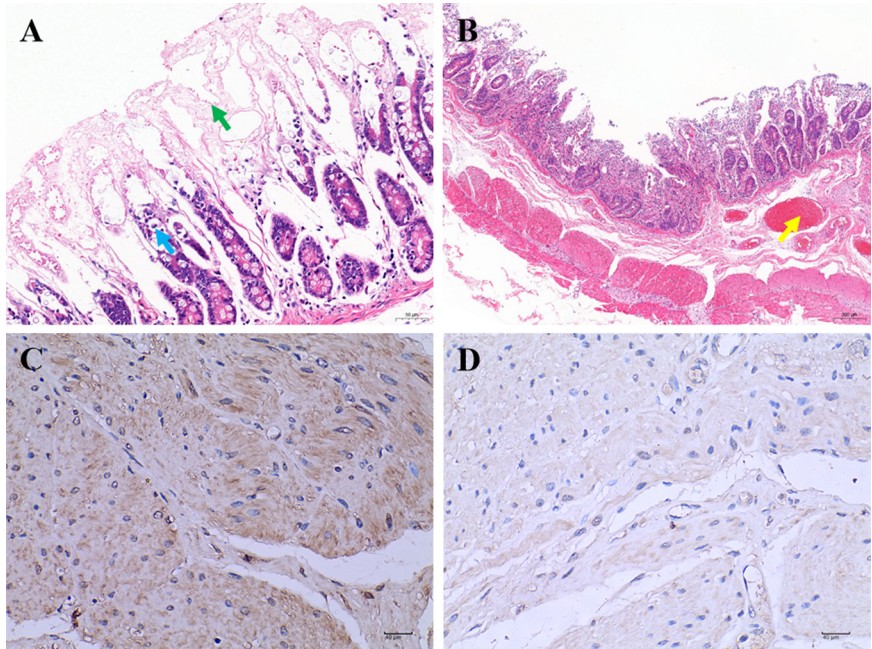

**FIG 5** Pathogenicity of AiV-D in yaks. (A) Hematoxylin and eosin (H&E) staining effects in duodenum after infection with AiV-D (×400 magnification): shedding of mucosal epithelial cells and necrosis of the intestinal glands, indicated with arrows (↑). Scale bar = 20 μm. (B) H&E staining effects in jejunum after infection with AiV-D (×400): hyperemia indicated with an arrow (↑). Scale bar = 20 μm. (C) AiV-D immunohistochemistry (IHC) expression indicated in brown (×400); IHC of the tissue was performed with anti-AiV-D polyclonal antibody (1/200) prepared in our laboratory. (D) Negative control for IHC.

7 dpi, with a peak at 6 dpi; after 7 dpi, the amount of virus shedding began to decrease until AiV-D was completely undetectable at 14 dpi (Fig. S10).

**Tissues distribution and re-isolation of AiV-D.** In addition, AiV-D was detected in the colon, lymph, jejunum, ileum, duodenum, cecum, and rectum of two infected yak calves (Table S4). Furthermore, we re-isolated AiV-D from positive tissue samples of infected yak calves using Vero cells.

## DISCUSSION

**Yak AiV-D strains may represent a novel genotype of AiV-D.** Three AiV-D isolates genomes have been identified in yaks. Four characteristics were observed in these novel strains: (i) 72.1% to 81.7% shared nt identity with known AiV-D strains; (ii) being grouped into the AiV-D species but forming an independent branch, as determined from the phylogenetic trees based on complete genomes and capsid protein P1 (VP0, VP3, and VP1 genes); (iii) VP0, VP3, and VP1 genes with unique aa changes, differing from those of known AiV-D strains (AiV-D1, AiV-D2, and novel AiV-D in sheep); and (iv) a different conformation of the 5′ UTR from that of known AiV-D strains. These characteristics suggest that the yak AiV-D strains differ from known AiV-D strains and likely represent a novel genotype in AiV-D.

**The novel AiV-D in yaks.** Because AiV-D is a newly emerging virus, knowledge of its epidemiology and molecular characteristics remains limited. Here, 24.8% (29/117) of diarrhea yak fecal samples were determined to be AiV-D–positive by qRT-PCR, and the virus was detected in all of the 7 farms in three provinces of China's Qinghai-Tibet Plateau; the two most distant farms were over 1,300 km away, suggesting a wide geographical distribution of AiV-D in yaks in China's Qinghai-Tibet Plateau. Previously, AiV-D in cattle had only been detected in Japan, with detection rates of 10.4% and 16.9% for AiV-D1 and AiV-D2, respectively, in 77 diarrhea fecal samples from black cattle in Kagoshima Prefecture, Japan (14). Yaks are a distinct breed of cattle on the Tibetan plateau, and calving occurs between March and May and calf diarrhea from May to August. Yaks are often kept in a semi-loose state because their feeding is managed in the form of grazing. Because local transportation on the Tibetan plateau is difficult and veterinary services are poor, it is difficult to collect

samples from yaks, especially disease material. Initially, this study was designed to monitor a batch of fecal samples from yaks with diarrhea. However, the samples showed no common diarrhea pathogens, and viral metagenomics were used to identify the virus species in the samples, leading to the unexpected discovery of AiV-D. After the presence of AiV-D in feces of yaks with diarrhea was confirmed, no more calf diarrhea samples could be collected that year; the following year (2022) was affected by the COVID-19 pandemic in China, which impeded normal sampling. This is the reason for the failure to collect any more calf diarrhea samples, and further work specifically for epidemiological survey is needed to improve this in the future. The samples used for AiV-D detection were obtained from three provinces in Sichuan, Qinghai, and Tibet on the Qinghai-Tibet Plateau, covering the main breeding areas of yaks. The detection rate of AiV-D was 24.8% and the farm positivity rate was 100.0%, the two furthest positive farms were more than 1,300 km apart, and the detection rate of AiV-D across the three provinces showed no clear difference. This research shows the first detection of AiV-D in fecal samples from diarrhea yaks, and further investigation revealed its wide geographical distribution in yaks in China's Qinghai-Tibet Plateau. These results help us understand the prevalence of AiV-D and indicate that this virus deserves more attention. Further investigations into its association with diarrhea are also warranted.

**AiV-D is a diarrhea-causing virus in yaks.** AiV-D is seen as an emerging virus in cattle and sheep detected in Japan and China (14, 15). Because of a paucity of epidemiological data and a lack of infectivity experiments, the biological importance of this virus still needs to be determined. In this study, AiV-D strains were successfully isolated, making it necessary to study its biological characteristics further. In the animal infection study, it was found that AiV-D caused diarrhea and intestinal bleeding in yak calves via oral inoculation. The virus was distributed in the jejunum, cecum, colon, ileum, duodenum, and rectum. There were similarities in the clinical manifestations and pathological changes in yaks infected with AiV-D to those in cattle infected with Aichivirus B (20). Despite the limited number of animals in this experiment, we demonstrated that AiV-D is a diarrhea-causing pathogen in yaks. There should be further investigation to better learn about the pathogenic characteristics of AiV-D in cattle and the host range of AiV-D.

**Molecular characterization of the VP0, VP3, and VP1 genes of AiV-D in yaks.** The P1 protein of *Kobuvirus* is mainly included in cellular receptor recognition, viral pathogenesis, and antigen diversity (21, 22). The core domain located from 195 to 253 aa in the VP0 gene of Aichivirus A is likely active in receptor binding and is associated with viral pathogenesis (22); these motifs were conserved in the same Aichivirus species and the VP0 gene exists in each representative Aichivirus species (15, 23, 24). Compared with all AiV-D strain sequences, 14 VP0 gene sequences in yaks had 7 unique aa changes in the RBD (200 to 261 aa). The VP3 of Aichivirus C inhibits the interferon (IFN)-$\beta$–triggered signaling pathway, thus playing an immune evasion role through the IFN signaling pathway (25). Compared with all AiV-D strain sequences, 14 VP3 gene sequences of yak AiV-D had 13 unique aa changes. Further study is necessary to explore the biological significance of unique aa changes in the VP0 and VP3 genes of AiV D in yaks.

The VP1 gene of Aichivirus A is suggested to play a role in viral pathogenesis (21), and residues 228 to 240 aa (PRPPPPLPPLPTP) in VP1 of Aichivirus A are a recognition motif for binding to the enteric receptor (22). Moreover, an antigenic epitope survey disclosed a high-antigenicity VP1 epitope at 21 to 43 aa (DNSPQPRTTFDYTDNPLPPDTKL), and this fragment is conserved in a range of Aichivirus strains from humans, pigs, cattle, canines, and sheep (26). In this study, the VP1 receptor-recognition motif sequences of the AiV-D identified in yaks (PRAPPTTASAPST) were found at 227 to 239 aa in VP1, the same as in AiV D2 genotype strain Kago-2-24. Compared with the other 2 known AiV-D genotype strains (Kago-1-22 and AB18), the yak AiV-D strains shared 2 identical aa substitutions with strain Kago-2-24 in VP1's potential RBD. Furthermore, in this study, we found that the antigenic epitope motif of AiV-D in yaks had 3 combining forms located at 22 to 44 aa in VP1 (VSAPETRTTFEYKDAPRPPDTML, ISEPETRTTFEFKDAPRPPDTML, and VSEPETRTTFEYKDAPRP PDTRL), which differs from the 3 known AiV D strains. The VP1 gene of Aichivirus is involved in neutralizing antibodies (26), and the aa changes in the antigenic peptide region in different

AiV-D strains likely influence the cross-protection of these antibodies, which can lead to immune escape for those strains. Therefore, there should be further attention on monitoring the variation of the AIV-D VP1 gene to offer a theoretical basis for the development of an effective vaccine.

**Conclusions.** In summary, the present study identified a novel genotype of AiV-D in yaks based on the characteristics of successfully isolated genomes. It also confirmed that this virus is a diarrhea-causing pathogen in yaks and has a wide geographical distribution in the Qinghai-Tibet Plateau. These findings expand both the host range of AiV-D and the pathogen spectrum of yaks and enhance our knowledge of the prevalence and diversity of AiV-D.

## MATERIALS AND METHODS

**Samples collection.** During May–August 2021, a total of 117 fecal samples were collected from diarrhea yaks in Sichuan (42 samples, Daofu county: N32°21′, E100°32′, 3 farms), Qinghai (38 samples, Hainan region: N34°38′, E98°55′, 2 farms), and Tibet (37 samples, Naqu region: N29°55′, 2 farms) in China (Table 1). In addition, each sample was transported on ice and stored at −80℃ in the lab.

**Viral nucleic acid extraction and deep sequencing.** A 20% fecal suspension in phosphate-buffered saline (PBS) was prepared and then clarified by centrifuging at $12,000 \times g$ for 15 min. Viral RNA was extracted from the suspension via RNAiso Plus (TaKaRa), and then reverse-transcribed using the PrimeScript RT reagent kit (TaKaRa) following the manufacturer's instructions. The obtained cDNA was stored at −20℃ to carry out further analysis.

The samples for deep sequencing were taken from 7 farms, except for 1 farm from which 3 samples were randomly selected, and the remaining 6 farms, from which 2 samples were randomly selected from each farm, for a total of 15 samples pooled for metagenomic analysis. In addition, viral RNA was extracted via the QIAamp Viral RNA Minikit (Qiagen), and reverse-transcribed using SuperScript III Reverse Transcriptase (Invitrogen); and the cDNA sample was sent to Novogene Biotech Co., Ltd. (Beijing, China) for NGS to determine the virus species in each sample.

**Screening for AiV-D in yak by qRT-PCR.** AiV-D was detected using the quantitative real-time RT-PCR (qRT-PCR) assay established in this study (AiV-D-F: 5′-CGCTGTCTGGAGAACCCTGAGTA-3′, AiV-D-R: 5′-GTTCGATGATACCACCAAGGAGC-3′; targeting the 3D gene; fragment length: 157 bp). This assay has good sensitivity and specificity, with a detection limit of 18.4 copies/$\mu$L (details of qRT-PCR are listed in the supplemental material). The PCR systems used are as follows: 12.5 $\mu$L TB Green Premix *Ex Taq* II (TaKaRa); 1 $\mu$L reverse primer (0.05 $\mu$M); 2 $\mu$L cDNA; 1 $\mu$L forward primer (0.05 $\mu$M); and 8.5 $\mu$L nuclease-free water. The PCR conditions used were as follows: 94℃ for 1 min, followed by 40 cycles at 95℃ for 15 s and 56℃ for 30 s; and a melt curve stage of 95℃ for 15 s and 60℃ for 1 min. The amplification products were sequenced (Sangon Biotech) in both directions for verification.

**Complete P1 region sequence amplification.** The P1 region contains the VP0, VP3, and VP1 genes. qRT-PCR was used to amplify the full VP0, VP3, and VP1 genes from AiV-D–positive yak diarrhea fecal samples. To amplify the complete VP0, VP3, and VP1 genes, we used the following primer pairs: 2F, 2R; 3F, 3R; 4F, 4R; 5F, 5R; and 6F, 6R (Table S1). The PCR products were purified via an Omega Gel kit (Omega), cloned to pMD19-T vector (TaKaRa Bio Inc., Japan), and sequenced (Sangon Biotech) in both directions for verification.

**Virus isolation and identification.** The AiV-D–positive sample supernatants were flask-cultured on Vero cells and incubated at 37℃ for 2 h; the mixtures were then discarded, and 4 mL of Dulbecco's modified Eagle's medium was added with 100 units/mL penicillin and 100 $\mu$g/mL streptomycin. The isolated strains were then purified via plaque technique when CPE was over 80% after three culture generations, during which virus titers were counted and expressed as the $TCID_{50}$. The procedures followed for indirect immunofluorescence (IF) assay and transmission electron microscopy were as previously described (27). As a note, an anti-KoV polyclonal antibody was prepared in our laboratory for IF.

**The genome amplification of AiV-D isolates in yak.** A total of 12 primer pairs (Table S1) were designed following the sequence contigs obtained through the high-throughput sequencing. The 5′ UTR and 3′ UTR of genomes were obtained using the Smart RACE kit (TaKaRa), and the obtained PCR products were cloned to a pMD19-T simple vector (TaKaRa) to perform sequencing.

**Sequence and phylogeny analyses.** The sequences were assembled using SeqMan software (version 7.0; DNASTAR Inc., WI, USA) and compared to other sequences available in GenBank using BLAST. The homologies of nucleotide and deduced amino acid sequences were determined using the MegAlign program in DNASTAR 7.0 software (DNASTAR Inc.). Phylogenetic trees based on nucleotide sequences were constructed using the maximum-likelihood method and the Kimura two-parameter model in MEGA version 7.0. Bootstrap analyses were performed based on 1,000 replicates.

**Experimental infection of animals with AiV-D isolates.** To understand the pathogenicity of AiV-D, the AiV-D isolate in this study was used to infect BALB/c mice and yak calves. A group of 10 four-day-old BALB/c mice were administered 200 $\mu$L of virus (virus titer: $10^7$ $TCID_{50}$/0.1 mL) orally, and 4 control BALB/c mice were administered 200 $\mu$L of supernatant from the Vero cells.

The 2-month-old healthy yak calves were purchased from a farm in Sichuan. The yak calves tested negative for bovine viral diarrhea virus, bovine rotavirus A, bovine nebovirus, bovine coronavirus, bovine norovirus, and AiV-D (28–32). The yak calves were randomly assigned to the infection and control groups (2 in the control group, 3 in the infection group). Yak calves were infected with 5 mL of virus (virus titer: $10^7$ $TCID_{50}$/0.1 mL) via oral inoculation. The control group was inoculated with 5 mL PBS. Anal swabs were collected from

yaks in each group at 1 to 7, 9, 11, and 14 dpi. The animals' clinical manifestations were identified, and infected animals with obvious clinical symptoms were euthanized to conduct further investigation.

**Distribution of AiV D in tissue from experimentally infected yak calves.** To explore virus distribution *in vivo*, experimentally infected yak calves were used to collect tissue samples from the heart, lung, kidney, liver, spleen, and intestine. In this study, qRT-PCR was used to detect the virus.

**Histopathological and immunohistochemistry observation.** Sections of 4% paraformaldehyde-fixed lung, spleen, heart, and intestinal tissues such as the jejunum, ileum, duodenum, cecum, colon, and rectum collected from experimentally infected yak calves were routinely processed and sent to Lilai Biotechnology Co., Ltd.(Chengdu, China) for H&E staining and immunohistochemistry (IHC). Notably, an anti-AiV-D polyclonal antibody was prepared in our laboratory for IHC.

**Ethical approval.** The testing program in this study was approved by the Animal Care and Use Committee of Southwest Minzu University, Chengdu, Sichuan, China. All animal procedures were performed according to the regulations and guidelines established by this committee and international standards for animal welfare.

**Data availability.** The AiV-D sequences described in this study have been deposited in GenBank under accession numbers OP776095 to OP776111.

## SUPPLEMENTAL MATERIAL

Supplemental material is available online only.
**SUPPLEMENTAL FILE 1**, PDF file, 2 MB.

## ACKNOWLEDGMENTS

This work was funded by the open competition program of the Top 10 Critical Priorities of Agricultural Science and Technology Innovation for the 14th Five-Year Plan of Guangdong Province (2022SDZG02); the Guangdong Modern Agro-industry Technology Research System (2022KJ114); the Special Fund for Scientific Innovation Strategy—Construction of High-Level Academy of Agriculture Science (202110TD, R2020PY-JC001); the Innovation Fund of Guangdong Academy of Agricultural Sciences—Industrial Special Project (202144); a project of the Collaborative Innovation Center of GDAAS (XTXM202202); Science and Technology Plan Projects of Guangdong Province (2021B1212050021); the National Key Research and Development Program of the 14th Five-Year Plan (grant no. 2021YFD1600203); and the Innovation Team for Emerging Animal Disease at Southwest Minzu University (2020NTD02).

The authors have declared no competing interests.

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
