## [Reviewer comments · Microbiology Spectrum]

Microbiology Spectrum

Isolation and characteristics of a novel Aichivirus D from yak

Nan Yan, Hua Yue, Quan Liu, Gang Wang, Cheng Tang, and Ming Liao

Corresponding Author(s): Ming Liao, South China Agricultural University College of Veterinary Medicine

Review Timeline:

Submission Date:	January 7, 2023
Editorial Decision:	January 31, 2023
Revision Received:	March 27, 2023
Accepted:	April 5, 2023

Editor: Biao He

Reviewer(s): The reviewers have opted to remain anonymous.

Transaction Report:

DOI: <https://doi.org/10.1128/spectrum.00099-23>

January 31, 2023

Prof. Ming Liao
South China Agricultural University College of Veterinary Medicine
College of Veterinary Medicine
483 Wushan Road
Guangzhou, Guangdong
China

Re: Spectrum00099-23 (**Isolation and characteristics of a novel Aichivirus D from yak**)

Dear Prof. Ming Liao:

Thank you for submitting this manuscript to Spectrum. The manuscript has been reviewed by two experts with expertise addressed in this study. As you can see below, both of them raised some concerns. Accordingly, our decision at this stage is "Modification."

Link Not Available

Sincerely,

Biao He

Journals Department
Reviewer comments:

Reviewer #1 (Public repository details (Required)):

Viral sequences are presented with Genbank submission numbers.

Reviewer #1 (Comments for the Author):

The study aims to characterize the genome of a novel Kobuvirus found in yaks. Furthermore, the study seeks to understand the prevalence of this virus in China and the pathogenesis in mice and yak.

The genetic characterization of the work is complete, refined, and well-described. However, the other study portions are not as refined and resemble data in a preliminary character. For example, the "prevalence" study is biased (diarrheic samples only). Furthermore, it does not provide any rationale for the sampling design and size of the population in question. Similarly, the pathogenesis study in yaks used a limited number of animals, and the design can be improved. For example, there is no fecal shedding data during the study (either virus titer or CT).

This reviewer also has concerns about the lack of details presented in the material and methods. The quality of the figures is not ideal. For example, it looks like an additional supplement was added to the cell culture media of the virus cells based on the color of the media in the figure. The FA definition is not good, and I am not sure about the IHC specificity based on the provided picture and the lack of negative control images.

Reviewer #2 (Comments for the Author):

In this manuscript, a novel genotype of AiV D from yaks was identified by Yan et al, and it was confirmed to be a diarrhea pathogen of yak calves by animal infection and the re-isolation of AiV D. Investigation of the prevalence and characteristics of AiV D was conducted in this study, and the results indicated that this novel AiV D strains have been widely prevalent in yaks from the Qinghai-Tibet Plateau in China, expanding the host range of AiV D and the pathogen spectrum of yaks. In general, abundant data was generated and presented in this manuscript, which might enhance the knowledge of the prevalence and diversity of AiV D. However, some concerns need to be addressed as below:

Major points:

1. Among the 117 diarrhea samples from yak, 29 were detected to be AiV D positive by RT-PCR (lines 97-98). However, complete P1 region sequences were obtained from the 14 AiV D positive samples (lines 103-104). How did you choose the 14 AiV D positive samples from all the 29 AiV D positive samples?
2. Lines 140-142, the sentence "The phylogenetic trees based on complete genomes and individual genes all showed that these strains were grouped into AiV D species, forming an independent branch (Fig.4, Supplementary Fig S6-S8)" is not correct, because the phylogenetic tree based on the complete nucleotide sequences of AiV L gene did not form an independent branch as showed in Fig S6. Additionally, the value of bootstrap under 60 is usually not displayed in phylogenetic tree.
3. Lines 164-165, authors described that "Infected yak calves showed watery diarrhea and depression at three dpi, and diarrhea was most severe at six dpi (Supplementary Fig.S10)". However, Supplementary Fig.S10 just showed the severe diarrhea at three dpi without six dpi. Viral shedding of infected yaks in different days post-infection by RT-qPCR is suggested to be added.
4. Lines 170-171, "AiV D was detected in the lymph, duodenum, jejunum, ileum, cecum, colon, and rectum of the two infected yak calves (Supplementary Table S3)", while in lines 166-167, "gross pathological changes were observed in the obtained duodenum and jejunum, characterized by intestinal bleeding (Supplementary Fig.S11)", how about the gross pathological changes in ileum, cecum, colon, and rectum?
5. Lines 264-265, "15 samples were selected from three provinces and pooled for metagenomics analysis as previously described". 15 samples were selected, Please explain the criteria.

Minor points:

1. There are many grammatical errors in this manuscript. The English should be well polished.
2. Abbreviations should be defined at first mention in the main body part, and then subsequently used throughout the manuscript, such as dpi and TCID50.
3. Line 122, "10-6.5, 10-6.8 and 10-7 TCID50/0.1mL" do authors mean "106.5, 106.8 and 107 TCID50/0.1mL"?
4. There are many errors in this manuscript that need to be corrected, such as line 143 "The", line 209 "caused cause", line 212 "naturally infected", line 272 "94 °C", line 299 "Genbank"...Please check the whole manuscript carefully.

Staff Comments:

Preparing Revision Guidelines

For complete guidelines on revision requirements, please see the journal Submission and Review Process requirements at

<https://journals.asm.org/journal/Spectrum/submission-review-process>. **Submissions of a paper that does not conform to Microbiology Spectrum guidelines will delay acceptance of your manuscript. "**

Please return the manuscript within 60 days; if you cannot complete the modification within this time period, please contact me. If you do not wish to modify the manuscript and prefer to submit it to another journal, please notify me of your decision immediately so that the manuscript may be formally withdrawn from consideration by Microbiology Spectrum.

In this manuscript, a novel genotype of AiV D from yaks was identified by Yan et al, and it was confirmed to be a diarrhea pathogen of yak calves by animal infection and the re-isolation of AiV D. Investigation of the prevalence and characteristics of AiV D was conducted in this study, and the results indicated that this novel AiV D strains have been widely prevalent in yaks from the Qinghai-Tibet Plateau in China, expanding the host range of AiV D and the pathogen spectrum of yaks. In general, abundant data was generated and presented in this manuscript, which might enhance the knowledge of the prevalence and diversity of AiV D. However, some concerns need to be addressed as below:

Major points:

1. Among the 117 diarrhea samples from yak, 29 were detected to be AiV D positive by RT-PCR (lines 97-98). However, complete P1 region sequences were obtained from the 14 AiV D positive samples (lines 103-104). How did you choose the 14 AiV D positive samples from all the 29 AiV D positive samples?
2. Lines 140-142, the sentence “The phylogenetic trees based on complete genomes and individual genes all showed that these strains were grouped into AiV D species, forming an independent branch (Fig.4, Supplementary Fig S6-S8)” is not correct, because the phylogenetic tree based on the complete nucleotide sequences of AiV L gene did not form an independent branch as showed in Fig S6. Additionally, the value of bootstrap under 60 is usually not displayed in phylogenetic tree.
3. Lines 164-165, authors described that “Infected yak calves showed watery diarrhea and depression at three dpi, and diarrhea was most severe at six dpi (Supplementary Fig.S10)”. However, Supplementary Fig.S10 just showed the severe diarrhea at three dpi without six dpi. Viral shedding of infected yaks in different days post-infection by RT-qPCR is suggested to be added.
4. Lines 170-171, “AiV D was detected in the lymph, duodenum, jejunum, ileum, cecum, colon, and rectum of the two infected yak calves (Supplementary Table S3)”, while in lines 166-167, “gross pathological changes were observed in the obtained duodenum and jejunum, characterized by intestinal bleeding (Supplementary

Fig.S11)”, how about the gross pathological changes in ileum, cecum, colon, and rectum?

5. Lines 264-265, “15 samples were selected from three provinces and pooled for metagenomics analysis as previously described”. 15 samples were selected, Please explain the criteria.

Minor points:

1. There are many grammatical errors in this manuscript. The English should be well polished.
2. Abbreviations should be defined at first mention in the main body part, and then subsequently used throughout the manuscript, such as dpi and TCID₅₀.
3. Line 122, “ $10^{-6.5}$, $10^{-6.8}$ and 10^{-7} TCID₅₀/0.1mL” do authors mean “ $10^{6.5}$, $10^{6.8}$ and 10^7 TCID₅₀/0.1mL”?
4. There are many errors in this manuscript that need to be corrected, such as line 143 “The”, line 209 “caused cause”, line 212 “naturally infected”, line 272 “94 °C”, line 299 “Genbank”...Please check the whole manuscript carefully.

Revision Note

Dear Editor/Reviewers

Thank you for kindly reviewing the manuscript entitled "Isolation and characteristics of a novel Aichivirus D from yak" (Spectrum00099-23). We are grateful for the professional suggestions of the reviewers. The manuscript has been revised accordingly, and revisions are marked in red in the revised manuscript.

Reviewer #1:

The study aims to characterize the genome of a novel Kobuvirus found in yaks. Furthermore, the study seeks to understand the prevalence of this virus in China and the pathogenesis in mice and yak.

The genetic characterization of the work is complete, refined, and well-described. However, the other study portions are not as refined and resemble data in a preliminary character. For example, the "prevalence" study is biased (diarrheic samples only). Furthermore, it does not provide any rationale for the sampling design and size of the population in question. Similarly, the pathogenesis study in yaks used a limited number of animals, and the design can be improved. For example, there is no fecal shedding data during the study (either virus titer or CT).

This reviewer also has concerns about the lack of details presented in the material and methods. The quality of the figures is not ideal. For example, it looks like an additional supplement was added to the cell culture media of the virus cells based on the color of the media in the figure. The FA definition is not good, and I am not sure about the IHC specificity based on the provided picture and the lack of negative control images.

Response: Thanks for your professional suggestions and the following responses to your comments.

(1). The epidemiological investigation in this research has certain limitations. Yaks are a distinct breed of cattle on the Tibetan plateau, with calving occurring between March and May and calf diarrhea from May to August. Yaks are often kept in a semi-loose state since the feeding management form of yaks as grazing. Because the local transportation on the Tibetan

plateau is very difficult and veterinary services are very poor, it is difficult to collect samples from yaks, especially yak disease material. Initially, this research was designed to monitor a batch of fecal samples from yaks with diarrhea. However, the samples showed no common diarrhea pathogens, and viral metagenomic was used to identify the virus species in the samples, leading to the unexpected discovery of AiV-D. After confirming the presence of AiV-D in the feces of the diarrhea yaks, no more calf diarrhea samples could be collected that year, and the subsequent year (2022) was affected by COVID-19 in China, which impeded the normal sampling. This is the reason for the failure to collect any more calf diarrhea samples, and further work specifically for the epidemiological survey is needed to improve this part in the future. The samples used for AiV-D detection were from three provinces in Sichuan, Qinghai and Tibet on the Qinghai-Tibet Plateau, covering the main breeding areas of yaks. The detection rate of AiV-D was 24.8% and the farm positivity rate was 100.0%, the two farthest positive farms were more than 1300 km apart, and the detection rate of AiV-D in the three provinces showed no clear difference. This result shows firstly AiV-D detection in fecal samples from diarrhea yaks, and further investigation revealed the wide geographical distribution of AiV-D in yaks in China's Qinghai-Tibet Plateau. In the revised manuscript, the relevant part of "prevalence" has been rewritten.

(2) . Since the identification of AiV-D in 2015, previous studies have only reported the detection of the virus in diarrhea feces, with no studies on virus isolation and pathogenicity. After successfully isolating the virus for the first time globally, we conducted a preliminary study on the pathogenicity of AiV-D. Despite the limited number of animals in this experiment, the results of clinical symptoms, histopathological changes and tissue distribution of the virus were demonstrated that AiV-D is a diarrhea pathogen in yaks. These findings both expand the host range of AiV-D and the pathogen spectrum of yaks. We will conduct further experiments to investigate the biological characteristics and detailed pathogenicity characterization of AiV-D. The fecal shedding of AiV-D in yaks was added in the revised manuscript.

(3) . We have added detailed information regarding the materials and methods in the revised manuscript, including the deep sequencing and the establishment of the qRT-PCR method. The methods details of qRT-PCR are listed in the supplementary materials.

(4) . We did not add additional supplements to the culture medium, probably due to the contrast of the cell pictures which did not look good, we have replaced the cell pictures with better quality. The anti-KoV polyclonal antibody was prepared in our laboratory for IF, and the anti-AiV-D polyclonal antibody was prepared in our laboratory for IHC, and the negative control for IHC was also added to Fig.2. The relevant description was added in the revised manuscript.

(5) . The English language within whole manuscript was revised carefully, we have used editing service to polish the English language of the revised manuscript.

EDITORIAL CERTIFICATE

This document certifies that the manuscript below was edited for correct English language usage, grammar, punctuation and spelling by qualified English-speaking editors at Proofine English Studio.

Paper Title:

Isolation and characteristics of a novel Aichivirus D from yak

Author:

Nan Yan

Date certificate issued:

27 Mar. 2023

Proofine English Studio, Beijing Zhi Bo Xuan Yu Cultural Development Ltd.
www.proofine.com
Unified Social Credit No. 91110208567461700L

Viral sequences are presented with Genbank submission numbers.

Response: The sequences of AiV-D described in this study has been deposited in GenBank under accession number OP776095-OP776111. The GenBank numbers of the three genome sequences are specifically noted in the revised manuscript, Yak/DF77 (GenBank No. OP776109), Yak/HN16 (GenBank No. OP776110) and Yak/NQX7 (GenBank No. OP776111).

Reviewer #2:

In this manuscript, a novel genotype of AiV D from yaks was identified by Yan et al, and it was confirmed to be a diarrhea pathogen of yak calves by animal infection and the re-isolation of AiV D. Investigation of the prevalence and characteristics of AiV D was conducted in this study, and the results indicated that this novel AiV D strains have been widely prevalent in yaks from the Qinghai-Tibet Plateau in China, expanding the host range of AiV D and the pathogen spectrum of yaks. In general, abundant data was generated and presented in this manuscript, which might enhance the knowledge of the prevalence and diversity of AiV D. However, some concerns need to be addressed as below:

Major points:

1. Among the 117 diarrhea samples from yak, 29 were detected to be AiV D positive by RT-PCR (lines 97-98). However, complete P1 region sequences were obtained from the 14 AiV D positive samples (lines 103-104). How did you choose the 14 AiV D positive samples from all the 29 AiV D positive samples?

Response: We are sorry that we did not describe clearly the background information of P1 amplification. In actuality, all of the 29 positive samples were used for P1 region amplification, and only 14 samples were amplified successfully from 29 AiV-D positive samples, and this is likely to be associated with the low virus amounts in the clinical samples. The relevant description was added in result in the revised manuscript.

2. Lines 140-142, the sentence "The phylogenetic trees based on complete genomes and individual genes all showed that these strains were grouped into AiV D species, forming an independent branch (Fig.4, Supplementary Fig S6-S8)" is not correct, because the phylogenetic tree based on the complete nucleotide sequences of AiV L gene did not form an independent branch as showed in Fig S6. Additionally, the value of bootstrap under 60 is usually not displayed in phylogenetic tree.

Response: Thank you for your suggestions, we have checked the phylogenetic tree and removed the wrong phylogenetic tree. The related sentences have been rewritten in the revised manuscript.

3. Lines 164-165, authors described that "Infected yak calves showed watery diarrhea and depression at three dpi, and diarrhea was most severe at six dpi (Supplementary Fig.S10)". However, Supplementary Fig.S10 just showed the severe diarrhea at three dpi without six dpi. Viral shedding of infected yaks in different days post-infection by RT-qPCR is suggested to be added.

Response: We are sorry for the incorrect description of the figure, the previous figure showed diarrhea at 6 dpi. We added 3 dpi and 6 dpi to the revised figure in Supplementary Fig.S7 in the revised manuscript. The fecal shedding of AiV-D in yaks was added in the revised manuscript, the methods details of qRT-PCR are listed in the supplementary materials.

4. Lines 170-171, "AiV D was detected in the lymph, duodenum, jejunum, ileum, cecum, colon, and rectum of the two infected yak calves (Supplementary Table S3)", while in lines 166-167, "gross pathological changes were observed in the obtained duodenum and jejunum, characterized by intestinal bleeding (Supplementary Fig.S11)", how about the gross pathological changes in ileum, cecum, colon, and rectum?

Response: The description of pathological changes and the histopathological changes in the ileum, cecum, colon, and rectum were added to the result in the revised manuscript.

5. Lines 264-265, "15 samples were selected from three provinces and pooled for metagenomics analysis as previously described". 15 samples were selected, Please explain the criteria.

Response: We are sorry that we did not describe clearly the criteria of selected samples for metagenomics analysis. The samples for deep sequencing were taken from 7 farms, except for 1 farm where 3 samples were randomly selected, and the remaining 6 farms where 2 samples were randomly selected from each farm, for 15 samples pooled for metagenomic analysis. The relevant description has been rewritten in the revised manuscript. The detailed number of samples in each farm was also added to Table 1.

Minor points:

1. There are many grammatical errors in this manuscript. The English should be well polished.

Response: Thanks for your suggestion, the English language within whole manuscript was revised carefully, we have used editing service to polish the English language of the revised manuscript.

EDITORIAL CERTIFICATE

This document certifies that the manuscript below was edited for correct English language usage, grammar, punctuation and spelling by qualified English-speaking editors at Proofine English Studio.

Paper Title:

Isolation and characteristics of a novel Aichivirus D from yak

Author:

Nan Yan

Date certificate issued:

27 Mar. 2023

Proofine English Studio, Beijing Zhi Bo Xuan Yu Cultural Development Ltd.
www.proofine.com
Unified Social Credit No. 91110208567461700L

2. Abbreviations should be defined at first mention in the main body part, and then subsequently used throughout the manuscript, such as dpi and TCID₅₀.

Response: Thanks for your suggestion, it was corrected in the revised manuscript.

3. Line 122, "10^{-6.5}, 10^{-6.8} and 10⁻⁷ TCID₅₀/0.1mL" do authors mean "10^{6.5}, 10^{6.8} and 10⁷ TCID₅₀/0.1mL"?

Response: It was corrected in the revised manuscript.

4. There are many errors in this manuscript that need to be corrected, such as line 143 "The", line 209 "caused cause", line 212 "naturally infected", line 272 "94 °C", line 299 "Genbank"...Please check the whole manuscript carefully.

Response: Sorry, the whole manuscript was revised carefully.

April 5, 2023

Prof. Ming Liao
South China Agricultural University College of Veterinary Medicine
College of Veterinary Medicine
483 Wushan Road
Guangzhou, Guangdong
China

Re: Spectrum00099-23R1 (**Isolation and characteristics of a novel Aichivirus D from yak**)

Dear Prof. Ming Liao:

It is a pleasure to let you know that your manuscript has been accepted, and I am forwarding it to the ASM Journals Department for publication. You will be notified when your proofs are ready to be viewed.

Sincerely,

Biao He
Editor, Microbiology Spectrum
